# *R2H*: Building Multimodal Navigation Helpers that *Respond to Help Requests*

**Yue Fan, Jing Gu, Kaizhi Zheng, Xin Eric Wang**

University of California, Santa Cruz

{yfan71, kzheng31, jgu110, xwang366}@ucsc.edu

## Abstract

Intelligent navigation-helper agents are critical as they can navigate users in unknown areas through environmental awareness and conversational ability, serving as potential accessibility tools for individuals with disabilities. In this work, we first introduce a novel benchmark, *Respond to Help Requests (R2H)*, to promote the development of multi-modal navigation helpers capable of responding to requests for help, utilizing existing dialog-based embodied datasets. R2H mainly includes two tasks: (1) Respond to Dialog History (RDH), which assesses the helper agent's ability to generate informative responses based on a given dialog history, and (2) Respond during Interaction (RdI), which evaluates the effectiveness and efficiency of the response during consistent cooperation with a task performer. Furthermore, we explore two approaches to construct the navigation-helper agent, including fine-tuning a novel task-oriented multi-modal response generation model that can see and respond, named *SeeRee*, and employing a multi-modal large language model in a zero-shot manner. Analysis of the task and method was conducted based on both automatic benchmarking and human evaluations. Project website: https://sites.google.com/view/response2helprequests/home.

## 1 Introduction

Assisting humans in real-world scenarios is a fundamental capability that AI agents should possess. An AI helper, communicating with humans in natural language based on visual observation in the environment and oracle information could significantly enhance work productivity and serve as an accessibility tool for individuals with disabilities. Figure 1 illustrates an example of a delivery person seeking assistance in navigating through an unfamiliar building. With the help of a navigation-helper agent, the delivery person can ask questions

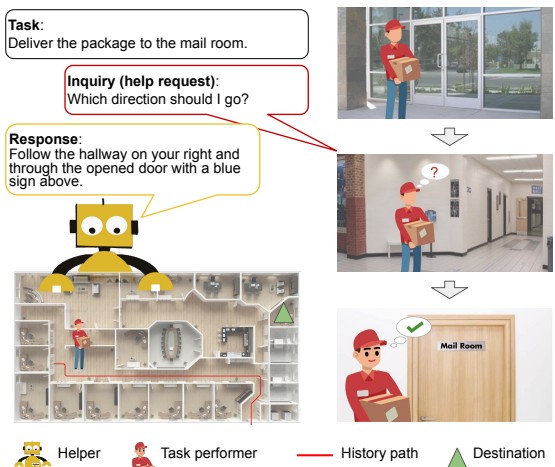

Figure 1: Example of a helper agent. A navigation helper provides responses to help a task performer who is delivering a package. The helper has access to oracle information that is not available to the task performer, such as the location of the destination and map of the environment.

about directions and receive responses that are tailored to visual information about the current surroundings.

Building such a helper agent poses significant challenges. It requires understanding the visual environment and the task performer's inquiries and leveraging oracle information to provide effective responses. Evaluating these agents also requires a task performer to show how well the helper agent performs in the real collaborative real scenario, where the task performer follows instructions and further sends inquiries when needed to the helper with the goal of completing tasks. Using a human as an oracle task performer would be the most intuitive setting, but it is impractical due to the high cost and low efficiency.

In this work, we introduce the Respond to Help Requests (R2H) benchmark, designed to automatically evaluate conversational multi-modal navigation helpers in a cooperative dynamic with another agent as the task performer. The R2H benchmark

incorporates pre-trained performer agents to follow the responses from the helper agent, and the helper agent's performance is then reflected in the performance of the fixed task performer. Leveraging three existing vision-and-dialog navigation datasets, CVDN (Thomason et al., 2020), AlFRED (Shridhar et al., 2020) and AVDN (Fan et al., 2022), our R2H benchmark introduces two novel tasks: the Respond to Dialog History task (RDH) and the Respond during Interaction task (RdI). In the RDH task, the helper agent generates a response to the inquiry in the dialog history from humans, aiming at facilitating the task completion of the performer agent. In the RdI task, the helper agent needs to generate multiple responses from the start of the navigation process with no prior dialog history till the task's success. To this end, R2H benchmark offers a pragmatic evaluation of the response from helper agents in both single- and multi-turn helper-performer cooperation.

We also present a multi-modal helper agent SeeRee for R2H benchmark, which leverages the oracle knowledge about the task and environment such as the destination location in a navigation task, to generate responses to inquiries from the task performer. SeeRee employs pre-trained vision and language models to handle multi-modal inputs. To manage long input sequences, SeeRee leverages a novel Conditional Optimized Sparse (COS) attention mask. Moreover, we introduce a Parse by Step, which leverages Large Language Model to transform ground-truth human responses into structured step-by-step navigation instructions. Those parsed instructions (instead of human responses) serve as a better training source with improved performance in helping the task performer. In experiments, SeeRee surpasses the baseline in generating effective responses and validates the COS attention mask and Parse by Step method. SeeRee's responses have been evaluated through human assessments, demonstrating high accuracy and a significant improvement in task success rate compared to the baseline.

Additionally, we ask human testers to rate the faithfulness and naturalness of the responses evaluate the response from helper agents with automatic scores. As a result, our experiments indicate that a higher language similarity to human helpers does not necessarily lead to a more successful conversational helper agent.

The main contributions are concluded as follows:

- We present the Respond to Help Requests (R2H) benchmark as a test-bed for automatically evaluating the capabilities of multi-modal conversational navigation-helper agent, that helps task performers complete tasks by providing natural language responses to inquiries based on environment information.

- We build two task helper agents, a novel task-oriented multi-modal helper agent, SeeRee, utilizing the Conditional Optimized Sparse (COS) attention mask and noise-free step-by-step instructions (Parse by Step) and a multi-modal LLM helper with mPLUG-Owl (Ye et al., 2023).

- Our experiments on the R2H benchmark and human evaluations of two helper agents over baseline also indicate that a closer linguistic resemblance to human helpers does not automatically translate into a more effective conversational helper agent.

## 2 The R2H Benchmark

Significant differences exist between building a helper agent and developing task performer agents for dialog-based multi-modal navigation tasks. Task performers, such as ones in CVDN (Thomason et al., 2020), DialFRED (Gao et al., 2022), and AVDN (Fan et al., 2022), are challenged as command followers, evaluated based on their performance of following dialog histories from human annotations, as shown in Figure 2i. However, the helper agent works as a supportive role, responds to questions from the task performer to facilitate task success. Therefore, the evaluation of the helper agent's performance could not be solely dependent on the agent itself, but on how the task performer benefited from the response. Therefore building the helper agent requires evaluations in a collaborative setting with task performers.

Involving humans as task performers to evaluate the helper agent is ideal but expensive. Alternatively, inspired by Padmakumar et al. (2022), Nguyen and Daumé III (2019) and Roman et al. (2020) that build helper and performer agents to collaborate as shown in Figure 2ii, we introduce the Respond to Help Requests (R2H) benchmark, involving task performer in the evaluation process as shown in Figure 2iii). In this way, the helper agent can be assessed comprehensively and realistically. R2H benchmark tests the agent's ability

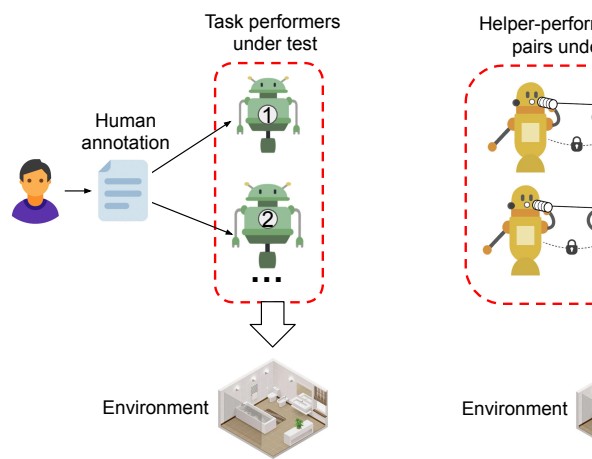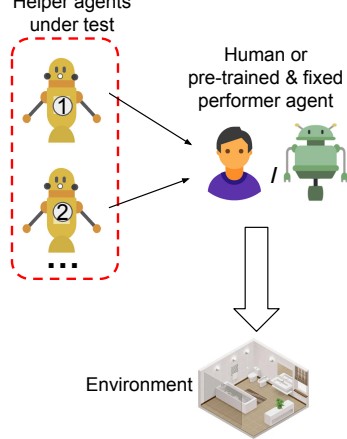

| (i) Benchmark for task performers. | (ii) Benchmark for helper-performer agent pairs. | (iii) Benchmark for helper agents. |

Figure 2: Comparison between different dialog-based embodied benchmark types. The benchmark either (i) evaluates task performers, where performers follow instructions in human annotations, (ii) evaluates helper-performer pairs, where the helper and performer agents need to learn and be evaluated together jointly, or (iii) evaluates helper agents only (our R2H benchmark), where helpers need to provide appropriate instructions according to performer and environment information.

to respond effectively in a wide range of scenarios, including two novel tasks, the Respond to Dialog History (RDH) task and the Respond during Interaction (RdI) task, building upon three existing vision-and-dialog navigation datasets. RDH task evaluates helper agents in a situation where partial human dialog history is provided and the RdI task aims at challenging the helper agents with real collaborative scenarios.

## 2.1 Respond to Dialog History Task

task, I think that is a stand alone task. Respond to Dialog History (RDH) Task focuses on evaluating the accuracy and completeness of the response from helper agents. The helper agent is challenged with understanding the dialog history and responds to help the task performer based on information about the task and environment in the form of image sequences. We developed environment-specific scripts to generate the image sequence which is introduced in Section 2.4. After the responses $\hat{r}_i$ generated, they will be concatenated with all available human dialog history $h_{i-1} = \{q_0, r_0, \ldots, q_{i-1}, r_{i-1}\}$ in the corresponding trajectory before the inquiries $q_i$ from human task performers. As a result, the generated response from the helper agent forms a new dialog history $\hat{h} = \{h_{i-1}, q_i, \hat{r}_i\}$ which becomes the input to the task performer.

## 2.2 Respond during Interaction Task

The Respond during Interaction (RdI) task challenges the ability of helper agents to cooperate consistently with the task performer. Similar to the RDH task, the RdI task involves a pre-trained task performer agent predicting navigation actions based on dialog histories. However, unlike the RDH task, no dialog history is initially provided in the RdI task, and the helper agent needs to respond to the navigation inquiries from the task performer constantly during the navigation. The task performer agent initiates inquiries $\hat{q}_i$ for help when needed and navigates based on the ongoing dialog between itself and the helper agent $\hat{h}_i = \{\hat{q}_0, \hat{r}_0, \ldots, \hat{q}_i, \hat{r}_i\}$, where $\hat{r}_i$ is the real-time responses from the helper agent to $\hat{q}_i$. The dialog $\hat{h}_i$ as a result of interaction serves as the primary source of guidance for the task performer agent, making the consistent quality of the helper agent's responses crucial for successful navigation. Additionally, since there is multi-turn helper-performer cooperation involved in the RdI task, the helper's communication efficiency can be evaluated by the number of conversation turns required for a fixed performer's outcome.

## 2.3 Task Performer Agent

Our R2H benchmark requires task performer agents to form helper-performer cooperation. In the RDH task, the task performer agent predicts navigation actions based on dialog history in specific

environments. As for the RdI task, the task performer also needs to generate navigation inquiries to accomplish the navigation task better.

R2H benchmark adopts state-of-the-art open-sourced task performer agents for vision-and-language navigation datasets. The task performer agent is pre-trained on the original training set with human dialogs $h_i$ including the response from the human helper $r_i$ and predicts actions based on $\hat{h}$ for completing the task. Therefore, the progress and success made by the task performer can be seen as a test of the accuracy and completeness of the responses generated by the helper agent.

## 2.4 Adapting Existing Datasets

R2H benchmark establishes the same tasks across different datasets.

**Datasets** R2H benchmark is built upon existing vision-and-dialog navigation datasets with dialogs between task performers and helpers.

- **CVDN** (Thomason et al., 2020) is situated in the Matterport3D simulator (Chang et al., 2017) with photo-realistic scenes. The dataset records the collaboration between the human task performer and the human helper to complete navigation tasks of finding target objects in different indoor environments.

- **DialFRED** (Gao et al., 2022) is built on Ai2-thor (Kolve et al., 2017) simulator with synthetic views. Similar to the CVDN, the helper and task performer collaborate to navigate to targets. However, each trajectory only corresponds to one pair of inquiry and response, making it unsuitable for RdI tasks.

- **AVDN** (Fan et al., 2022) is an aerial vision-and-language dataset that includes dialogs, navigation trajectories, and visual observation between the helper and task performer. The dataset is annotated upon a continuous state photo-realistic drone simulator where the goal of the task performer is to control the drone in the simulator with a top-down view to navigate to a certain destination.

**Environment-specific Adaption** Given the variability of available information across different datasets and environments, the R2H benchmark is designed with a harmonizing approach, converting the environment-specific information into image

sequences. Script-based samplers are designed for each environment to generate the image sequences by leveraging oracle information. The sampler outputs image sequences showing the task performer's views on the shortest path to the destination. Especially for the CVDN dataset, following the data collection process, a connectivity graph for viewpoints is used to generate the shortest path and therefore the image sequence length is variable but limited to views within 5 viewpoints of the current position. Examples are shown in the appendix. For the AVDN dataset, since the allocentric direction description, such as "turn south" could be involved, we keep the image sequence all oriented with the north at the top and indicate the drone direction with a red arrow. As a result, helper agents can be evaluated in different datasets with the same input format, which enhanced the benchmark's versatility to adapt further to new datasets.

## 2.5 Metrics

Since we aim to evaluate how capable the response generated by helper agents is in helping the task performer, we adopt the primary metrics for task completion from each dataset: Goal Progress (GP) in CVDN evaluates the distance of the progress made towards the destination, where it is computed as the trajectory length, deducted by the remaining trajectory from the current location to the destination viewpoint; Success Rate (SR) in DialFRED, shows the ratio of tasks being completed successfully; Success weighted by inverse Path Length (SPL) (Anderson et al., 2018) as the dominant metric in AVDN measures the Success Rate weighted by the total length of the navigation trajectory.

## 3 Models

### 3.1 SeeRee

In this section, we introduce the helper agent that can *see and respond*, named SeeRee. SeeRee generates responses to task-oriented navigation inquiries from the task performer. As is illustrated in Figure 3, our helper agent SeeRee generates natural language responses to the task performer's inquiries based on the task and environment information that is not aware by the task performer. The image sequences are padded to a fixed length (Lin et al., 2022), encoded by Video Swin Transformer (Liu et al., 2022) and then concatenated with BERT text embeddings (Devlin et al., 2019). Finally, the embeddings are fed into a multi-modal transformer

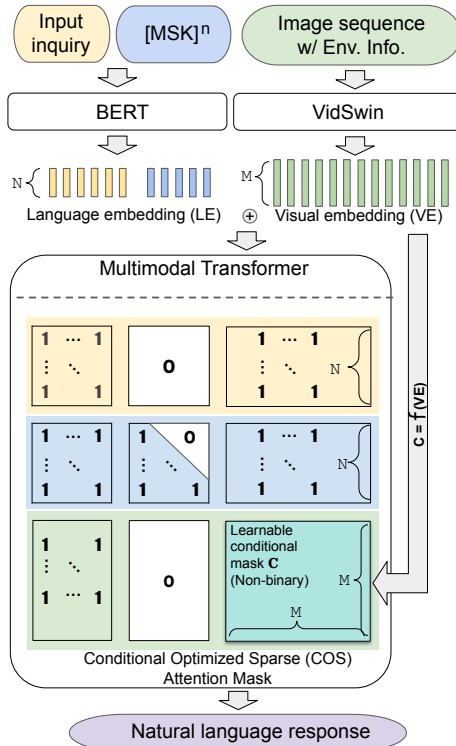

Figure 3: Overview of SeeRee. The visual and text inputs are encoded and fed into the multi-modal transformer, where Conditional Optimized Sparse (COS) attention mask is applied. The COS attention mask has fixed binary values except for the learnable conditional mask $C$ for visual embedding (VE) that is conditioned on VE itself. Yellow, blue, and green colored rows correspond to the attention masks for LE of input inquiry and generated response and VE, respectively.

which generates the natural language response in an auto-regressive way. SeeRee is trained end-to-end with Mask Language Modeling (Devlin et al., 2019) and please refer to the appendix for the training detail.

### 3.1.1 Multi-modal Transformer

Following prior multi-modal language generation studies (Hu et al., 2020; Lin et al., 2022), our multi-modal transformer takes as input the embedding containing both text and image information and generates natural language responses in a unidirectional sequence-to-sequence generation process. We treat the input inquiries as prompts for generating the response, and a special token [CLS] is added to the end of the inquiry.

At inference time, the text is generated in an auto-regressive manner, where we insert multiple [MSK] tokens after the [CLS] token and predict tokens to replace [MSK]tokens one by one unidirectionally until the prediction is [EOS] or all the [MSK] tokens are predicted.

### 3.1.2 Conditional Optimized Sparse (COS) Attention Mask

One challenge in generating responses for dialog-based embodied tasks is effectively modeling the long input image sequence, reducing the redundancy in the repetitive images but keeping the critical details. To this end, we introduce a Conditional Optimized Sparse (COS) attention mask for the multi-modal transformer, as shown in Figure 3. The mask can be divided into three row sections, corresponding to in what range the embeddings for input inquiry, the response generated, input image sequence can attend to, respectively. The first row section shows that the language embedding (LE) for input inquiry can attend to itself and the input visual embedding (VE). The second row section means the LE for the generated response can attend to the VE and all the previous LE, allowing unidirectional text generation (Devlin et al., 2019). The third row section indicates that the VE can attend to partial itself and the LE of input inquiry. Especially, instead of being fulling binary and pre-defined, a learnable conditional mask $C$ that is non-binary and sparse is adopted in the third row section of the COS attention mask, controlling the self-attention of the VE. $C$ is conditioned on VE, and the mapping is modeled by:

$$C = \sigma(f(\text{VE})), \tag{1}$$

where $\sigma(x) = \frac{1}{1+e^{-x}}$ and $f$ is a multi-layer perceptron. In this way, our COS attention mask uses a conditional optimizing strategy that optimizes the attention mask based on the image sequence. As a result, COS attention mask enables better encoding and understanding of long visual input and improves the response generation result.

### 3.1.3 Response Preprocessing with Parse by Step

Training a helper agent to imitate the responses in the human dialog directly may not be optimal as they may be unorganized and include utterances irrelevant to the task. Structured step-by-step instructions are easier for the helper agent to learn. Therefore, inspired by the idea of in-context learning (Brown et al., 2020; Kojima et al.), we propose a Parse by Step method that prompts GPT-3 (Brown et al., 2020) to preprocess the ground-truth responses in the training data. We detail the designed prompts and examples in the appendix. Af-

| Helper | CVDN | | DialFRED | | AVDN | |
|---|---|---|---|---|---|---|
| | Seen GP ↑ | Unseen GP ↑ | Seen SR ↑ | Unseen SR ↑ | Seen SPL ↑ | Unseen SPL ↑ |
| Human Annotator | 6.9 | 5.1 | 49.1 | 33.4 | 14.7 | 16.5 |
| RMM $\mathcal{G}$ | 4.7 | 2.8 | 46.5 | 32.1 | 2.4 | 3.6 |
| Multimodal LLM | 5.3 | 3.6 | 47.0 | **33.8** | 0.7 | 1.5 |
| SeeRee | **6.5** | **4.9** | **49.1** | 33.1 | **4.6** | **4.4** |

Table 1: Results of Respond to Dialog History (RDH) task on CVDN, DialFRED, and AVDN dataset (Thomason et al., 2020; Gao et al., 2022; Fan et al., 2022). We replace the original response in validation sets with responses from different helper agents. With a fixed task performer agent, a better performance of the performer agent represents a more effective response from the helper agent.

ter Prase by Step, the preprocessed training data is parsed in a step-by-step manner with a streamlined language pattern. As a result, the learning objectives of SeeRee is the preprocessed human response $Y = P(R)$, where $P$ is the Parse by Step method, and $R$ is the original human response in the dialog of the training set.

## 3.2 Multi-modal Large Language Model

Besides our SeeRee model, we introduce another navigation-helper agent constructed from a multi-modal large language model (LLM). Specifically, we employ mPLUG-Owl (Ye et al., 2023), a State-Of-The-Art multi-modal LLM that is able to take as input an image sequence and text for language generation. mPLUG-Owl is originally trained with a large amount of uni-modal and multi-modal data in two stage paradigm including instruction tuning. To leverage the extensive knowledge that mPLUG-Owl accumulated through training and avoid the potential issue associated with limited task-specific training data, we adopt mPLUG-Owl in a zero-shot manner. The build of a helper agent based on mPLUG-Owl serves as a valuable comparison for SeeRee and sheds light on the potential of leveraging LLMs in building navigation-helper agents.

## 4 Experiments

## 4.1 Setup

We initialize the encoders and multi-modal transformers in SeeRee with weights from SwinBert (Lin et al., 2022) to benefit from its image sequence understanding ability and then finetune SeeRee on the training set of each dataset separately. We adopt the RMM Guide model ($\mathcal{G}$) (Roman et al., 2020) as the baseline model in our experiments. Please refer to the appendix for more implementation details.

## 4.2 RDH Task

**Task Performer Agents** For RDH task on CVDN dataset (Thomason et al., 2020), we use HAMT[1] (Chen et al., 2021), pre-trained on the original Navigation from Dialog History (NDH) task in CVDN as the task performer agent. For AVDN (Fan et al., 2022) and DialFRED (Gao et al., 2022) datasets, we leverage the task performer in the original work, i.e., HAA-transformer and DialFRED model [2]. We further detail the task performers in the appendix.

**Result** As indicated in Table 1, the task performer agent attains the best overall performance across all three datasets when guided by responses generated by the SeeRee helper agent. On both CVDN and DiaFRED datasets, SeeRee demonstrates performance levels that are strikingly similar to those of human helpers. This is likely attributable to the fact that the environments used in the CVDN are visually akin to the data employed during the pre-training phase of SeeRee's encoders. Such similarity facilitates efficient fine-tuning and leads to heightened performance. In addition, for the DialFRED dataset, due to the predominance of template-based human utterances, the challenge in response generation is simplified. Meanwhile, the AVDN dataset is more challenging due to the drone simulator's adoption of a continuous control space, and complex visual information leads to difficulty in reaching the precise destination. Still, SeeRee outperforms the baseline by a large margin. We present results from more LLM baselines and case studies of the RDH task in Appendix E and C.

## 4.3 RdI Task

**Task Performer Agents** We utilize the task performer agent from the pioneering work RMM (Roman et al., 2020) deployed on CVDN dataset (Thomason et al., 2020), which combines two components: the LSTM-based RMM Navigator model ($\mathcal{N}$), responsible for predicting navigation actions, and RMM Questioner model ($\mathcal{Q}$), generating inquires to the helper agent after every four navigation actions. Despite its comparatively simple structure, the RMM task performer stands out as the sole task performer developed that possesses both language and navigation capabilities: generating navigation actions and crafting natural language

---

[1]https://github.com/cshizhe/VLN-HAMT.

[2]Due to unavailable model weights, we reproduced the model with comparable results as in the original paper.

| Helper | COS attention mask | Parse by Step | Seen Validation | | | Unseen Validation | | |
|---|---|---|---|---|---|---|---|---|
| | | | GP | B2 | R | GP | B2 | R |
| Human Annotator | - | ✗ | 6.9 | - | - | 5.1 | - | - |
| | - | ✓ | 7.3 | - | - | 5.1 | - | - |
| SeeRee | ✗ | ✗ | 5.4 | 9.9 | 21.8 | 4.7 | 10.0 | 22.1 |
| | ✓ | ✗ | 5.9 | **14.4** | **25.5** | 4.7 | **13.9** | **25.5** |
| | ✓ | ✓ | **6.5** | 13.8 | 24.2 | **4.9** | 13.2 | 24.1 |

Table 2: Ablation study of the SeeRee agent on CVDN dataset (Thomason et al., 2020) based on RDH task GP and language generation metrics, BLUE2 (B2) and ROUGE-L (R). We directly apply our Parse by Step method to the original validation data created by humans, showing that Parse by Step maintains essential task-related information. The results show the effectiveness of COS attention mask and Parse by Step in task-oriented response generation.

navigation inquiries. Future works for building such language- and navigation-capable task performers in AVDN and DialFRED environments are needed. Furthermore, given the flexibility of our system, should a superior model be developed, the task performer could be effortlessly swapped.

**Result** The RdI task is conducted with a maximum 20 turns of conversation between the task performer and helper on the unseen validation set of the CVDN dataset. As depicted in Figure 4, we plot the mean GP of the task performer agent concerning the dialog turns happened during the navigation. As the dialog turn between helper and performer increases chronologically, more information is provided to the task performer. The navigation stops once the task performer reaches the maximum steps, 20, or a stop action is generated where the goal progress remains constant thereafter. Based on the result, the multi-modal LLM as the helper agent facilitates the most effective performance from the task performer at the end of the maximum conversation turns. However, it's worth noting that the response from SeeRee is shown to be less noisy on its effect and therefore is potentially more reliable over extended dialogue turns whereas multi-modal LLM tends to generate responses with a more varied effectiveness. Moreover, with less than half the maximum dialogue turns, SeeRee's responses yield a superior result that is 90% close to the navigation result at the end of the maximum turn. This underscores SeeRee's greater communication efficiency.

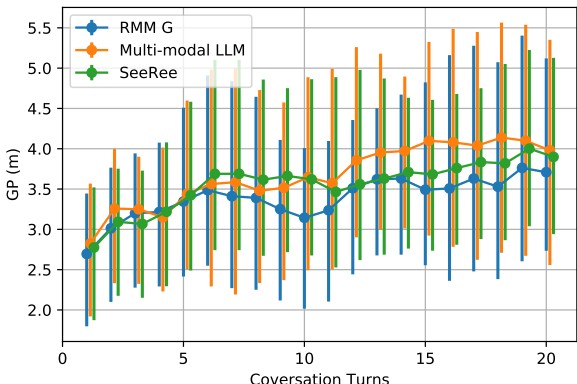

Figure 4: Results of Respond during Interaction (RdI) task on CVDN dataset (Thomason et al., 2020). The mean Goal Progress (GP) of the same task performer collaborating with different helper agents is plotted with respect to the number of conversation turns happened. Error bars show the relative range of the GP. Multi-modal LLM enables a better but noisier performance of the task performer than SeeRee.

| Helper | Task Performer | Task Completion | Subjective Response Evaluation | |
|---|---|---|---|---|
| | | GP ↑ | Naturalness ↑ | Faithfulness ↑ |
| No helper | | 3.54 | - | - |
| RMM $\mathcal{G}$ | Human | 6.32 | 73/100 | 59/100 |
| Multi-modal LLM | | 8.44 | **75/100** | 60/100 |
| SeeRee | | **9.89** | 72/100 | **75/100** |

Table 3: Results of human evaluation on RdI task on the CVDN (Thomason et al., 2020). The human tester plays the task performer role via an interface interacting with the helper agents. Task completion and subjective response evaluation are collected. The response from SeeRee is most effective despite being less natural.

## 4.4 Ablation Study

Based on CVDN dataset (Thomason et al., 2020), we conduct two ablation studies to explore the Parse by Step method and COS attention mask for SeeRee. We first let the task performer infers on the human dialog from the original dataset, but the response within is processed by our Parse by Step method. Then we create ablated SeeRee model by repeating the same training on SeeRee model twice, but incrementally replacing a naive fixed attention mask with the COS attention mask and applying our Parse by Step method.

**Response Effectiveness Result** The first ablation study that evaluate of Parse by Step method on the original validation set serves as a sanity check. As the result in Table 2, it shows that human responses processed with Pase by Step keep the information contained in the original response and they are overall equally capable as the original response. Addi-

tionally in the second ablation study, the GP of the same task performer cooperated with different ablated SeeRee shows that both COS attention mask and Parse by Step leads to major improvements to the effectiveness of the response in facilitating task success.

**Language Generation Similarity Result** As shown in Table 2, we also evaluate the ablated models with language generation metrics, BLUE2 (Papineni et al., 2002) and ROUGE-L (Lin and Och, 2004). BLUE2 score and ROUGH-L score drops when Parse by Stop method is applied, but GP in the RDH task receives a major increase. This indicates that a high language similarity to human responses does not necessarily equate to a more effective conversational helper agent.

## 4.5 Human Evaluation

To further evaluate the performance of helper agents, we conduct human evaluations based on the RdI task. Human participants act as task performers navigating 60 randomly selected trajectories in validation sets of CVDN (Thomason et al., 2020). During the simulation, participants can control their movement in the environment using their keyboards and ask questions to the helper agent whenever needed. We evaluate both settings where the helper agent exists or not. Human participants are also asked to rate the naturalness and faithfulness of the response from the helper. The average GP and subjective rating of the responses are shown in Table 3. The result shows that SeeRee provides the best results in terms of task completion. Through subjective evaluation, we find that SeeRee achieves significantly higher scores in terms of response faithfulness. Despite being trained with data preprocessed by Parse by Step, where the training supervision is no longer the original human utterances, it still achieves a close score of naturalness compared to the baseline model trained on original human responses. Through these evaluations, we show the ability of SeeRee to help human users complete embodied tasks.

## 5 Related Work

**Dialog-based Multi-modal Embodied Benchmarks** Previous dialog-based multi-modal embodied benchmarks usually focus on evaluating either task performers (Thomason et al., 2020; Gao et al., 2022; Shi et al., 2022; Gu et al., 2022) or the corresponding pairs of task performer and

helper (Roman et al., 2020; Hahn et al., 2020; Padmakumar et al., 2022). For instance, the CVDN (Thomason et al., 2020) evaluates a task performer to navigate to a desired room by dialog histories. Gao et al. (2022) developed a dialogue-enabled embodied instruction following benchmark, Dial-FRED, based on the ALFRED benchmark (Shridhar et al., 2020) and presented a task performer framework. Further, there is a wide range of activities studied in these tasks, such as navigating to a specific location (Roman et al., 2020), locating positions (Hahn et al., 2020), and interacting objects (Padmakumar et al., 2022). Compared to these benchmarks, our R2H benchmark aims for better helper agents and is the only benchmark for sole helper evaluation.

**Multimodal-based Language Generation** Building helper agents is in line with a growing collection of methods applied to the visual question-answering task. A unified vision-and-language framework can be trained to handle a variety of tasks, including the question-answering problem, using a single objective (Cho et al., 2021; Wang et al., 2022). Fu et al. (2021) tackled the video question-answering problem by adopting a video transformer to model the temporal dynamics of video inputs explicitly. One problem shared by these works is the input frames to the model are limited in quantity, whereas the helper agent has to take a long image sequence as input to include adequate information. Lin et al. (2022) developed a learnable sparse attention mask for video caption that enables long frame sequence as input. However, the learned sparse attention mask is fixed after training, lacking generalization ability compared to the COS attention mask of SeeRee.

## 6 Conclusion

In this paper, we introduce the Respond to Help Requests (R2H) benchmark with two tasks, Respond to Dialog History (RDH) and Respond during Interaction (RdI), assessing a helper agent's guidance capabilities. With the R2H benchmark, we build and evaluate two navigation-helper agents, SeeRee and Multi-modal LLM model. The results show that they both outperformed the baseline model. Through further ablation study and human evaluation, we prove the effectiveness of SeeRee in assisting humans with tasks and argue that the effectiveness of responses cannot be determined by their linguistic resemblance to human dialogue alone.

## Limitations

R2H benchmark presents a platform for helper agents that create natural language to address queries from task performers. Yet, the assessment of such an helper agent mandates a capable task performer agent that can not only navigate but also communicate, given that the efficacy of the helper can be gauged through final task fulfillment, and the task performer shouldn't act as a constraint. Also, the complexity of the real world surpasses that of a simulated environment, thereby imposing additional prerequisites for the task performer. Furthermore, the lack of abundant dialog-based embodied datasets also restricts the progression of both performer and helper agents.

## Ethics Statement

The human evaluation part of this project is classified as exempt by Human Subject Committee vis IRB protocols. Additionally, we recognize the potential ethical issues related to training language generation models using human dialog data, such as the possibility of the model learning harmful language without understanding its context. To mitigate this concern, our proposed data preprocessing method, Parse by Step, converts the responses in the training data into more structured and task-specific instructions, effectively reducing the presence of noise and profanity in the training data. As a result, the likelihood of our model generating inappropriate responses is greatly minimized.

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

# A Implementation Details

## A.1 Helper Model

**SeeRee** For the training of SeeRee, we apply Mask Language Modeling (MLM) (Devlin et al., 2019) to the response where $80\%$ tokens are masked with [MSK] tokens, and $10\%$ tokens are changed randomly. Cross entropy loss is applied to predictions for the masked tokens:

$$L_{MLM} = \Sigma_i L_{CrossEntropy}(y_i, \hat{y}_i), \quad (2)$$

where $y_i$ is the masked token at position $i$ and $\hat{y}_i$ is the prediction. Additionally, in order to let the COS attention $c$ mask attend to the specific details of the Visual Embedding (VE) that are most relevant to the task, we enforce $C$ to be sparse, letting the VE to sparsely attend to itself using a sparsity loss (Lin et al., 2022):

$$L_{SPARSE} = \lambda \times \sum_{i=1}^{M} \sum_{j=1}^{M} |C_{i,j}|, \quad (3)$$

where $\lambda$ is a regularization hyperparameter and $C_{i,j}$ is the value of the learnable conditional mask $C$.

SeeRee is trained on CVDN (Thomason et al., 2020), DialFRED (Gao et al., 2022) and AVDN (Fan et al., 2022) datasets individually using AdamW optimizer (Loshchilov and Hutter, 2018) for 20k iterations with a batch size of 6 and learning rate of $1e^{-4}$. The training data used are converted from the original training set from each dataset, with the environment-specific scripted sampler that generates image sequences with oracle environment information. We select the trained weights based on the RDH task evaluation result. Training takes about 12 hours on one NVIDIA A6000 GPU.

**Multi-modal Large Language Model** We utilized mPLUG-Owl (Ye et al., 2023) as a representative method for Large Language Model in this paper. mPLUG-Owl takes images and text as input and output natural language. It achieved state-of-the-art performance in various multi-modality tasks. Providing the further path in the form of images and a proper prompt, mPLUG-Owl directly outputs the guidance for the performer agent. Table 4 shows prompt templates. The QUESTION is relevant to the current task, and it is not strictly formed. For example, it could be "Should I continue forward?" in CVDN, "What does the object look like?" for DialFRED, "I am on top of a building block, Can I see the destination?" for AVDN.

**RMM $\mathcal{G}$** RMM $\mathcal{G}$ is an LSTM-based model that generates natural language response based on the input image sequence and dialog history. We train RMM $\mathcal{G}$ from scratch with the same data used for fine-tuning SeeRee, using a batch size of 8 and a learning rate of $1e^{-4}$.

## A.2 Task Performer Agents

We aim to leverage the best available task performers in R2H benchmark.

For CVDN dataset, we select History Aware Multimodal Transformer (HAMT) (Chen et al., 2021) as the task performer agent in RDH task. HAMT model is designed for Vision-and-Language Navigation (VLN) tasks. The HAMT model incorporates a long-horizon history into multi-modal decision making. It efficiently encodes all past panoramic observations via a hierarchical vision transformer and then combines text, history, and current observation to predict the next action. The model is first trained end-to-end using several proxy tasks, including single-step action prediction and spatial relation prediction, and then reinforcement learning is used to improve the navigation policy further. As the time we conduct the experiment, HAMT has achieved state-of-the-art results on a broad range of VLN tasks, including CVDN (Thomason et al., 2020). However, HAMT model is only capable for navigation action prediction and cannot generate questions for asking help. Therefore, we adopt RMM Navigator model ($\mathcal{N}$) with RMM Questioner model ($\mathcal{Q}$) (Roman et al., 2020) in RdI task, which is the only task performer model to the best of our knowledge that is designed to navigate while communicating in natural language. Both RMM $\mathcal{N}$ and $\mathcal{Q}$ are lstm based models and the RMM Questioner model is designed to generate questions at a fixed interval. It takes into account the Navigator's perspective in the environment and the dialog history.

For AVDN and DialFRED datasets (Fan et al., 2022; Gao et al., 2022), we use HAA-Transformer and DialFRED model proposed along with the dataset as the task performer agent because they are still leading the leaderboard [3] [4]. DialDRED model and HAA-Transformer model are both based on Episodic Transformer model (Pashevich et al., 2021). The DialFRED model trained on DialFRED

---

[3]https://eval.ai/web/challenges/challenge-page/1859/overview

[4]https://eval.ai/web/challenges/challenge-page/2049/overview

| | |
|---|---|
| **Prompt for CVDN:** | According to the given images, which way to go? Please only tell me the moving direction. Do not introduce the room details. $QUESTION |
| **Prompt for DialFRED:** | The images first show a sequence of first person view of a robot. Based on the image sequence, please answer the question about $QUESTION |
| **Prompt for AVDN:** | The images first show the drone view and red arrow shows current direction of the drone. How could the drone reach the destination in green bounding box? Note, north is up. Give instruction to the drone on $QUESTION? Concisely describe landmarks but do not mention red arrow and green box. |

Table 4: Text prompts for mPLUG-Owl. In RDH task, $QUESTION is the human inquires from the validation set. In RdI task, $QUESTION is generated by task performer agent in real-time.

| | |
|---|---|
| **Prompt for CVDN:** | Commander says: 'Yes to the kitchen. Go to the left of the fireplace and then all the way up the stairs.' Step by step: 1. Yes. 2. go to the kitchen 3. go the left of the fireplace, 4. go upstairs. Commander says:____ Step by step: 1. |
| **Prompt for AVDN:** | Commander says: 'Hi drone, if you go four o'clock, and go to a small building, go forward to the bigger building, your destination.' Step by step: 1. Go four o'clock. 2. fly to small building. 3. destination is the bigger building. Commander says: 'Hey drone, if fly over the white building.' Step by step: 1. fly over the white building. Commander says:____. Step by step: 1. |

| Dataset | Original response | Processed response |
|---|---|---|
| CVDN | Yeah keep going around the outside till you get to the end. And sorry about the mixup at first. | 1. Yeah. 2. Keep going around the outside. 3. Get to the end. |
| AVDN | Hey drone, head southwest, go through a small vegetation and arrive at a house that is the final destination. | 1. head southwest. 2. go through a small vegetation. 3. final destination is a house. |

Table 5: Examples of our Parse by Step method on the CVDN dataset (Thomason et al., 2020) and AVDN dataset (Fan et al., 2022). We fill the original response to the blank of the prompt and input to GPT-3 for sentence completion. The output from the GPT-3 becomes the processed response with mainly task-related instructions kept and organized in steps. Through Parse by Step, we preprocess the response in the training set.

dataset focuses on three types of questions: location clarification, appearance clarification, and direction clarification. The HAA-Transformer model trained on the AVDN dataset can predict both navigation waypoints and human attention.

## A.3 Prompts and Examples for Parse by Step

We design different prompts for applying Parse by Step on CVDN dataset (Thomason et al., 2020) and AVDN dataset (Fan et al., 2022). Table 5 shows the designed prompt and some example preprocessed results. To process the training set responses, we first insert the original response into the blank of the prompt, which is then inputted into GPT-3 for sentence completion. The output from GPT-3, primarily retaining task-related instructions arranged in steps, is subsequently treated as the processed response. Furthermore, we also eliminate the item numbers to streamline the instructions further. This approach aids in preserving the essential task-related information while enhancing the response's readability and conciseness.

## B Additional Benchmark Analysis

We conduct additional analyses on our R2H benchmark across the three datasets as shown in Table 6. Especially, the datasets we have adopted cover a wide range of environment types, including both indoor (CVDN and DialFRED) and outdoor (AVDN) settings, as well as synthetic (DialFRED) and photo-realistic environments (CVDN and AVDN). This diversity adds depth and robustness to our benchmark.

## C RDH Task Examples

In this section, we show examples of our RDH task on all three datasets, CVDN dataset (Thomason et al., 2020), DialFRED dataset (Gao et al., 2022) and AVDN dataset (Fan et al., 2022). As shown in Figure 8, 7 and 6 each example includes the input inquiry from human, the sampled input image sequence from the environment oracle information, the response generated by SeeRee and the human ground truth response.

## D Results of RDH Task on More Metrics

Table 1 shows the performance of RDH task on the main metric. Here Table 8 exhibits the results on more metrics for a thorough performance understanding.

Goal Progress (GP) evaluates the distance of the progress made towards the destination. Success

| | # navigation trajectories | # querries per trajectory | Avg length of human response (words) | Avg length of human query (words) | Avg # images in input image sequence of helper agents |
|---|---|---|---|---|---|
| CVDN | 2050 | 2.3 | 14.9 | 9.7 | 11.8 |
| DialFRED | 120958 | 1.0 | 6.0 | 7.0 | 10.8 |
| AVDN | 5372 | 1.8 | 19.7 | 9.0 | 9.1 |

Table 6: Statistic analysis on R2H benchmark across the three datasets. The columns from left to right are the number of navigation trajectories; the number of queries per trajectory, which is the same as the number of responses per trajectory; the average length of responses and queries provided by human annotators and the average number of images in the image sequences sampled from the environment by script-based samplers, which serve as input to the helper agent.

Rate (SR) shows the ratio of tasks being completed successfully. Success weighted by inverse Path Length (SPL) and Path Weighted Success Rate (PWSR) measure the Success Rate weighted by the total length of the navigation trajectory. Here we could draw similar conclusions with Table 1. Note that AVDN dataset (Fan et al., 2022) is more challenging due to the drone simulator's adoption of a continuous control space and complex visual information. The mistakes made by the helper are likely to be amplified if the task performer misses the target location and overshoots, or if the initial direction provided by helper agent is inaccurate. Even though all methods have a negative impact on the goal progress, SeeRee minimized the influence while achieving the highest Success Rate.

## E    Additional LLM Baselines

Comprehensive baselines are essential for validating our approach, particularly in light of the growing focus on Large Language Models (LLMs). Given the complexity of our tasks, the multimodal LLM (mPLUG-Owl) serves as an adequate LLM baseline as it inherently supports our task requirements by natively accommodating both image sequences and text for language generation. Additionally, we have also explored two additional baselines that leverage the power of LLMs as detailed below and the result is shown in Table 7.

**Multimodal-LLM + ChatGPT**    We experiment with prompting the Multimodal-LLM (mPLUG-Owl) to generate only captions for the image sequences. Subsequently, we use ChatGPT to serve as the helper agent, generating responses based on the captions and queries from the task performer.

**BLIP2 Model with stacked images input**    We employ the BLIP2 model (Li et al., 2023) in a zero-shot manner. BLIP2 model benefits from a generic and efficient pre-training strategy that combines pretrained vision models with LLMs for vision-language pretraining. Due to the model's limitation of accepting only a single image input, we stack the input image sequences into a single image arranged in four columns. The text prompt used is the same as the prompt for Multimodal-LLM (mPLUG-Owl) as shown in Table 4.

## F    Human Evaluation Details

In our human evaluation study, we recruited five students from the university and paid with at least $17/h as the human task performer. They consented to participate and contribute their data used for this work. These participants were given instructions on how to use the simulator before the evaluation began and had the same prior knowledge about the project. The participants are randomly assigned a navigation-helper agent and CVDN data including the initial instruction, starting, and target location. An interface, as shown in Figure 5, is built where an image box will show the image returned from the simulator and a text box allows both navigation control commands and sending natural language inquires for helper agents. It enables an easy interaction among the human task performer, helper agent, and simulator so that the participants can experience a real-time collaboration between helper and performer with only a screen and keyboard. Upon completion of the task, we assess task success and goal progress in the same way as in Thomason et al. (2020) and we ask the participant to rate the naturalness and faithfulness of the response from the helper in the last task session.

| | CVDN validation seen GP | CVDN validation unseen GP | DialFRED validation seen SR | DialFRED validation unseen SR | AVDN validation seen SPL | AVDN validation unseen SPL |
|---|---|---|---|---|---|---|
| Multimodal LLM baseline (mPLUG-Owl) | 5.3 | **3.6** | **47.0** | **33.8** | 0.7 | 1.5 |
| mPLUG-Owl + ChatGPT | 5.3 | 3.5 | 43.5 | 31.8 | 1.2 | 1.6 |
| BLIP2 | **5.6** | 3.4 | 38.6 | 24.7 | **3.8** | **5.0** |

Table 7: Evaluation of multimodal LLM baselines on the RdH task. Additional LLM baselines (bottom two rows) are compared with regard to the multimodal LLM baseline using mPLUG-Owl.

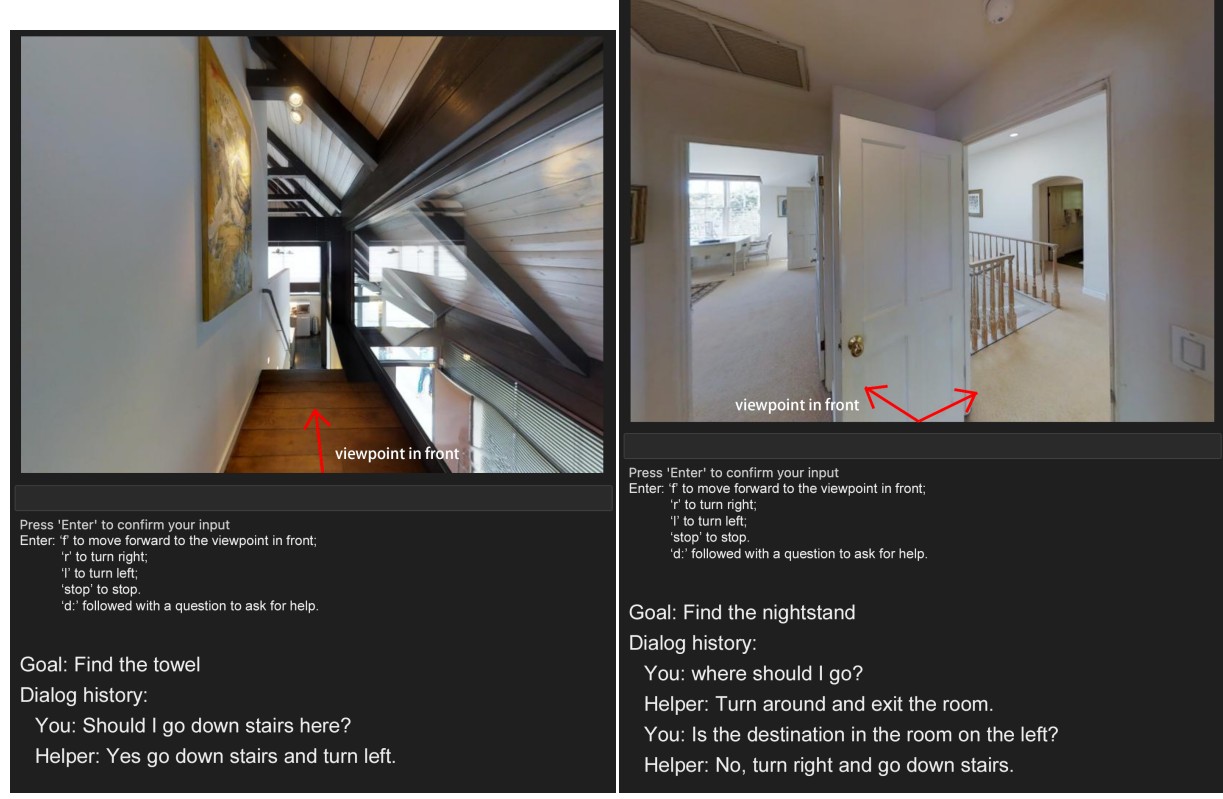

(i) Example 1                              (ii) Example 2

Figure 5: Examples of the interface used during human evaluation based on RdI task with CVDN (Thomason et al., 2020) dataset.

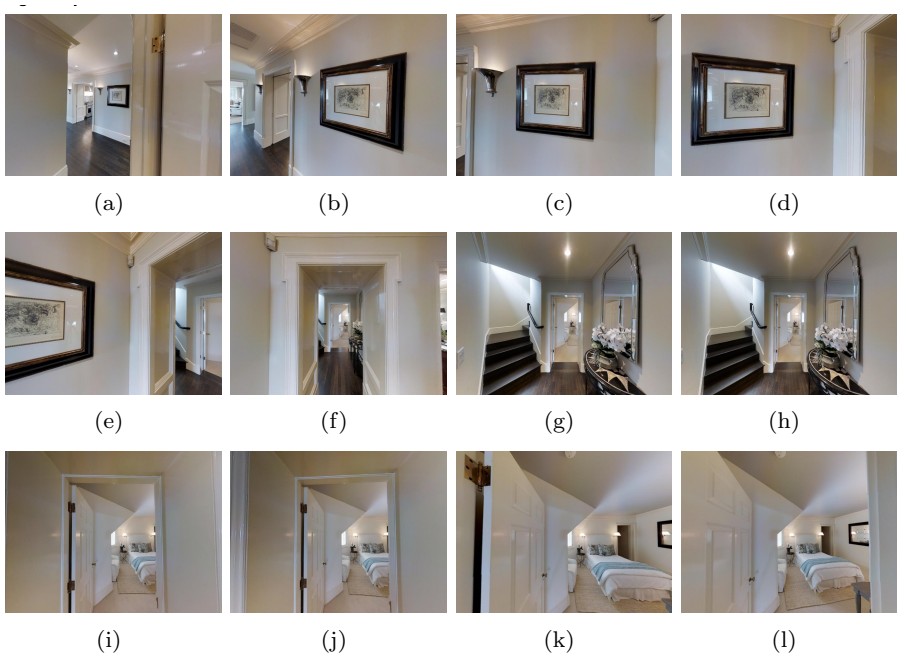

Figure 6: An example of data in RDH task on CVDN dataset.
Human inquiry: Shall I continue assuming this is not the bathroom? Ground truth human response: Yes leave the bedroom turn right go into the bedroom on the left.

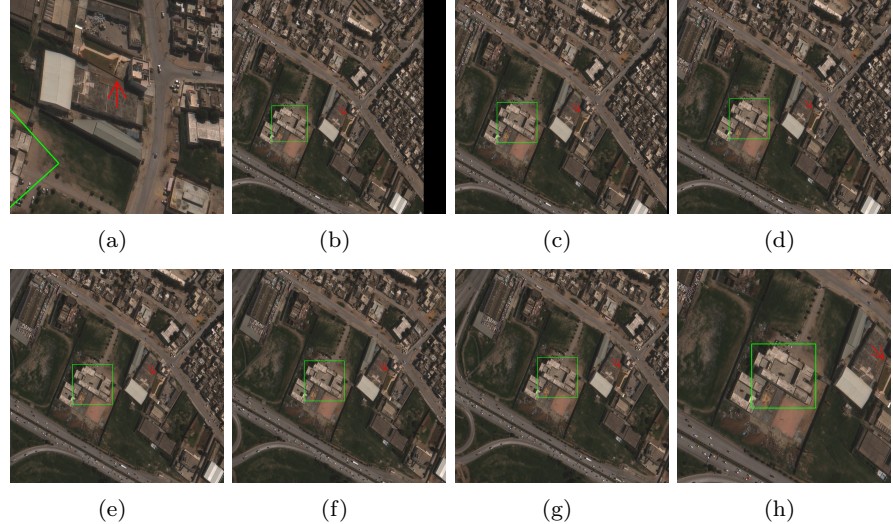

Figure 7: An example of data in RDH task on the AVDN dataset.
Human inquiry: I see some warehouses in my view. Am I near the destination? How to go to destination?
Ground truth human response: You are not close to your destination yet, head north and you will reach the complex gray warehouse office.

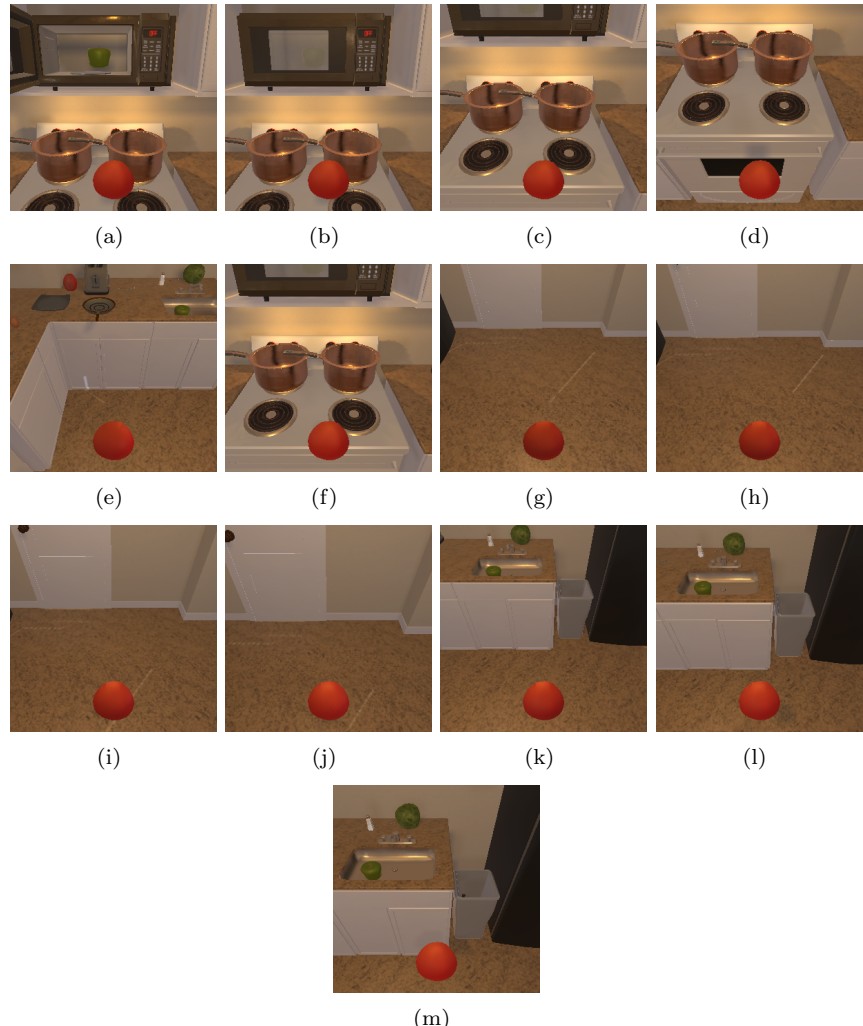

Figure 8: An example of data in RDH task on DialFRED dataset.
Human inquiry: What is the color of the sink basin? Ground truth human response: The sink basin is Grey.

| Helper | Task Performer | Seen Validation | | | Unseen Validation | | |
|---|---|---|---|---|---|---|---|
| | | GP ↑ | SPL ↑ | SR ↑ | GP ↑ | SPL ↑ | SR ↑ |
| Human Annotator | | 6.9 | 17.6 | 20.7 | 5.1 | 11.2 | 17.1 |
| RMM $\mathcal{G}$ | HAMT | 4.7 | 8.6 | 14.4 | 2.8 | 6.8 | 10.7 |
| Multimodal LLM | | 5.3 | **14.7** | 17.5 | 3.6 | 8.2 | 13.1 |
| SeeRee | | **6.5** | 14.0 | **17.8** | **4.9** | **10.1** | **15.2** |

(a) CVDN dataset

| Helper | Task Performer | Seen Validation | | Unseen Validation | |
|---|---|---|---|---|---|
| | | SR ↑ | PWSR ↑ | SR ↑ | PWSR ↑ |
| Human Annotator | | 49.1 | 39.1 | 33.4 | 14.6 |
| RMM $\mathcal{G}$ | DialFRED Model | 46.5 | 35.3 | 32.1 | 13.8 |
| Multimodal LLM | | 47.0 | 32.5 | **33.8** | 14.2 |
| SeeRee | | **49.1** | **39.1** | 33.1 | **14.3** |

(b) DialFRED dataset

| Helper | Task Performer | Seen Validation | | | Unseen Validation | | |
|---|---|---|---|---|---|---|---|
| | | GP ↑ | SPL ↑ | SR ↑ | GP ↑ | SPL ↑ | SR ↑ |
| Human Annotator | | 56.3 | 14.7 | 17.3 | 55.2 | 16.5 | 20.4 |
| RMM $\mathcal{G}$ | HAA-Transformer | -37.0 | 2.4 | 3.5 | -26.6 | 3.6 | 4.9 |
| Multimodal LLM | | -96.5 | 0.7 | 0.8 | -95.9 | 1.5 | 2.2 |
| SeeRee | | **-29.6** | **4.6** | **5.7** | **-24.9** | **4.4** | **6.1** |

(c) AVDN dataset

Table 8: Results of Respond to Dialog History (RDH) task on CVDN dataset (a), DialFRED dataset (b) and AVDN dataset (Thomason et al., 2020; Gao et al., 2022; Fan et al., 2022). We replace the response in validation sets with responses from different helper agents. Since the task performer agent is trained with human dialog and is fixed, it relies on the information contained in the response to complete the task. Thus, the better the task performer agent performs in the task, the more effective the response is.