# OpenReview forum: "R2H: Building Multimodal Navigation Helpers that Respond to Help Requests"
_EMNLP/2023/Conference — EMNLP 2023 Main_

### Official Review · Reviewer_6i43 · 2023-08-03

**Soundness:** 3

**Excitement:**

3: Ambivalent: It has merits (e.g., it reports state-of-the-art results, the idea is nice), but there are key weaknesses (e.g., it describes incremental work), and it can significantly benefit from another round of revision. However, I won't object to accepting it if my co-reviewers champion it.

**Paper Topic And Main Contributions:**

This paper presents the Respond to Help (R2H) benchmark as a testbed for automatically evaluating the capabilities of multi-modal conversational navigation helper agent. The helper agent helps task performers complete tasks by providing natural language responses to inquiries based on environment information.
The benchmark comprises two tasks: (1) Respond to Dialog History (RDH), which evaluates the helper agent's ability to generate informative responses from given dialog histories, and (2) Respond during Interaction (RdI), which assesses the effectiveness and efficiency of the agent's responses during consistent cooperation with a task performer. The paper explores two approaches to constructing the helper agent: (1) fine-tuning a task-oriented multi-modal response generation model capable of seeing and responding, and (2) employing a multi-modal large language model in a zero-shot manner.

**Questions For The Authors:**

Have you investigated whether different task performers affect the results? Appendix A.2 says the use of different task performers for different datasets. It would be helpful to see the results of using different task performers within the same dataset.

**Reasons To Accept:**

1. The paper is well-written and presents the information clearly.
2. The Conditional Optimized Sparse attention masks for reducing redundancy in repetitive images while retaining critical details is intuitive and effective.

**Reasons To Reject:**

1. The main concern lies in the motivation behind the R2H benchmark. It only evaluates the helper agent, while some datasets require collaboration between both helper and performer agents (Fig 2 ii). The explanation provided in Line 161-164 is not sufficiently convincing.
2. The paper needs more benchmark analysis. While the benchmark is built on three existing datasets, readers need more details to understand its intricacies fully.
3. The paper discusses the Language Generation Similarity Result in Line 529-532: a high language similarity to human responses may not necessarily indicate a more effective conversational helper agent. Further analysis is required to understand the underlying reasons for this situation.
4. More analysis of Conditional Optimized Sparse attention mask is needed to prove its effectiveness, such as what does the learnable conditional mask C capture, and how does it appear on the test set.

**Reproducibility:**

4: Could mostly reproduce the results, but there may be some variation because of sample variance or minor variations in their interpretation of the protocol or method.

**Reviewer Confidence:**

4: Quite sure. I tried to check the important points carefully. It's unlikely, though conceivable, that I missed something that should affect my ratings.

---

> ### Author Rebuttal · Authors · 2023-08-29
>
> Thank you for your thoughtful review comments. In the following response, we have organized our answers to address each point raised in your comments. We are committed to incorporating additional results in the future revision of the paper based on your feedback.
>
>
> **The unique focus of our R2H benchmark: building helpers**
>
> Our work focuses on the development and importance of multimodal navigation helper agents in real-world, practical scenarios. While much of the existing literature has been centered on improving task performers either independently (Figure 2 i) or in a two-agent collaboration setting (Figure 2 ii), there is substantial untapped potential in specifically advancing helper agents for collaborative tasks. First, creating helper agents is a valuable research endeavor in its own right, with real-world applications such as guiding humans in package delivery or room-finding tasks. Second, even in the two-agent collaboration setting, optimizing the helper agent is an under-explored direction, and our study showcases how these optimized helper agents can be readily integrated into two-agent collaboration, thereby enhancing the effectiveness of the task performers.
>
> Moreover, existing dialog-based embodied datasets like CVDN, DiaFRED, and AVDN offer a unique opportunity. These datasets are compiled through human-to-human interactions where humans perform both the helper and performer roles. Consequently, these datasets are not exclusively geared towards evaluating two-agent settings (Figure 2 ii). They also serve as robust platforms for the assessment of either task-performing agents (Figure 2 i) or helper agents (Figure 2 iii) in isolation. Leveraging the rich human-to-human interactions captured in these datasets, we have established a new benchmark tailored for the evaluation of helper agents in scenarios where the task performer is either a fixed model or a human participant.
>
> In sum, our work not only distinguishes itself by focusing on independent helper agents but also pioneers new pathways for research and practical applications in this relatively unexplored domain.
>
> **Additional benchmark analysis**
>
> We appreciate the reviewer's call for a more detailed benchmark analysis. To address this, we have conducted additional analyses on our R2H benchmark across the three datasets included in the following table. Specifically, we calculated:
>
> -   The number of navigation trajectories;
>
> -   The number of queries per trajectory, which is the same as the number of responses per trajectory;
>
> -   The average length of responses and queries provided by human annotators;
>
> -   The average number of images in the image sequences sampled from the environment by script-based samplers, which serve as input to the helper agent.
>
> |          | # navigation trajectories | # querries per trajectory | Avg length of human response (words) | Avg length of human query (words) | Avg # images in the input image sequence to helper agents |
> |----------|---------------------------|---------------------------|--------------------------------------|-----------------------------------|-----------------------------------------------------------|
> | CVDN     | 2050                      | 2.3                       | 14.9                                 | 9.7                               | 11.8                                                      |
> | DialFRED | 120958                    | 1.0                       | 6.0                                  | 7.0                               | 10.8                                                      |
> | AVDN     | 5372                      | 1.8                       | 19.7                                 | 9.0                               | 9.1                                                       |
>
>
> We would like to emphasize that the datasets we've adopted cover a wide range of environment types, including both indoor (CVDN and DialFRED) and outdoor (AVDN) settings, as well as synthetic (DialFRED) and photo-realistic environments (CVDN and AVDN). This diversity adds depth and robustness to our benchmark.
>
> For further clarity, we have provided examples from each dataset in Figure 6, 7 and 8 of our paper's Appendix, allowing readers to fully grasp the intricacies of our benchmark.
>
>
>
>
> **Language similarity result and effectiveness of the conversational helper agents**
>
>
>
> First, we want to point out that a single query from the task performer could elicit multiple valid responses, and there is no single "ground truth" response for every given query. For instance, if a generated response closely resembles a human response but reverses a crucial directional word (e.g., "left" is replaced with "right"), the effectiveness of the response for aiding the task performer in navigation would be compromised.
>
>
>
> Besides the intuitive analysis, we also draw conclusions from quantitative results. We use BLUE2 and ROUGE-L to evaluate the similarity of generated responses compared with human responses, and the GP to measure how well the task performer moves towards the destination, indicating the effectiveness of the response in aiding the task performer. In Table 2 of our paper (the bottom two rows), we present evidence that language similarity metrics, BLUE2 and ROUGE-L, do not consistently correlate with the GP metric. For instance, when BLUE2 decreases from 14.4 to 13.8 and ROUGE-L decreases from 25.5 to 24.2, the GP actually increases from 5.9 to 6.5. This quantitative result underscores that a high language similarity to human responses is not a reliable indicator of an effective conversational helper agent.
>
>
>
> To further substantiate this point, we conducted an experiment where we deliberately altered human responses in the CVDN dataset. Specifically, we removed the directional words "left" and "right" and also eliminated the last two words in all responses, which usually indicate the target. We compare the evaluation of these altered human responses with responses generated from our SeeRee helper in the table below. Despite achieving high language similarity scores, the altered human responses lead to significantly lower task performer's GP scores. demonstrating that language similarity is not a reliable indicator of the effectiveness of a conversational helper agent.
> |                         | Seen Validation GP | Seen Validation BLUE2 | Seen Validation ROUGE-L | Unseen Validation GP | Unseen Validation BLUE2 | Unseen Validation ROUGE-L |
> |-------------------------|--------------------|-----------------------|-------------------------|----------------------|-------------------------|---------------------------|
> | Altered human responses | 6.2               | **83.6**              | **95.6**                | 4.2                 | **83.5**                | **95.9**                  |
> | SeeRee                 | **6.5**            | 13.8                  | 24.2                    | **4.9**              | 13.2                    | 24.1                      |
>
>
>
>
> **Conditional Optimized Sparse attention mask**
>
> Firstly, we would like to point out the ablation study result based on CVDN dataset presented in Table 2, showing that the COS attention mask enhances the ability of the helper model to assist the task performer in reaching the destination. Besides the CVDN dataset, we also conducted an ablation study on the COS attention mask using the AVDN dataset. Same as the ablation study in Section 4.4 and Table 2, we train the SeeRee with a fixed COS attention mask (all ones) on the AVDN dataset, and the evaluation is done with the same task performer as used in the paper, the HAA-Transformer. We list the results in the table below.
>
> |         helper                      | Seen validation SPL | Unseen validation SPL |
> |-------------------------------|---------------------|-----------------------|
> | SeeRee w/o COS attention mask | 3.18                | 3.88                  |
> | SeeRee                        | 4.6                 | 4.44                  |
>
> The result shows quantitative evidence as in the paper, underscoring the effectiveness of our proposed COS attention mask.
>
>
>
> To further understand the behavior of the COS mask, we conducted a qualitative analysis through visualization of the COS attention mask. We observe that the COS mask learns a hybrid strategy. For different input image sequences, there is a generally similar mask pattern. However, adjustments are made, particularly when the length of the input image sequence varies. These variations between COS masks for different sequence lengths are quite significant. Moreover, we randomly sampled a few examples in the validation set of AVDN, and for each token in the video embedding, we computed the averaged corresponding value in the COS attention mask. Then since the video embedding has dimensions T/2 * H/32 * W/32, we reshaped the corresponding COS attention mask and observed that the learned COS attention mask shows a preference for the edge regions of every image. We believe this is because, in the CVDN dataset, the edge regions of the input images are more coherent and thus more informative for determining the direction of movement.
>
>
>
> **Different task performers within the same dataset**
>
> In the paper, we select HAA-Transformer as the task performer for AVDN dataset. Here we ran additional experiments using a different task performer, HAA-LSTM, within the same AVDN dataset to evaluate helper agents on the RDH task. The new results on AVDN unseen validation set are shown in the following table.
> |       Helper         | SPL of HAA-LSTM  | SPL of HAA-Transformer   |
> |----------------|--------------------------|----------------------------|
> | RMM G          | 2.8                    | 3.4                       |
> | Multimodal LLM | 3.6                    | 3.7                       |
> | SeeRee         | 4.9                     | 4.4                       |
>
>
>
> Our new results show that the evaluation outcome is consistent with different task performers. This consistency reinforces the robustness of our approach, indicating that it is not heavily influenced by the choice of task performer.

---

### Official Review · Reviewer_DxYF · 2023-08-06

**Soundness:** 3

**Excitement:**

4: Strong: This paper deepens the understanding of some phenomenon or lowers the barriers to an existing research direction.

**Paper Topic And Main Contributions:**

The paper introduces a new benchmark called Respond to Help (R2H), aimed at facilitating the development of multi-modal navigation assistants. This benchmark leverages existing dialog-based embodied datasets to encourage the creation of navigation helpers capable of responding effectively to help requests. R2H consists of two tasks: Respond to Dialog History (RDH), assessing the agent's capacity to provide informative responses based on past dialog, and Respond during Interaction (RdI), which gauges response effectiveness during cooperative interactions. Two strategies for constructing the navigation helper agent are explored: fine-tuning a task-oriented multi-modal model called SeeRee, designed for perceiving and responding, and using a large multi-modal language model in a zero-shot manner. The research entails both automatic benchmarking and human evaluations to analyze the tasks and methods. The authors plan to release the code and data associated with this study.




**Reasons To Accept:**

1. The paper addresses the fundamental capability of AI agents to assist humans in real-world scenarios. This is an essential and practical area of research, as AI systems that can communicate naturally with humans and provide assistance based on visual observations hold the potential to significantly enhance productivity and accessibility, particularly for individuals with disabilities.
2. The authors introduce the Respond to Help (R2H) benchmark, which serves as a crucial contribution to the field. This benchmark is designed to automatically evaluate conversational multi-modal navigation helper agents in dynamic cooperative interactions with other agents acting as task performers.
3. The paper introduces a novel Conditional Optimized Sparse (COS) attention mask and a Parse by Step approach to transform ground-truth human responses into structured, noise-free step-by-step instructions. These contributions enhance the training process and improve the performance of the navigation helper agent.

**Reasons To Reject:**

The experimental evaluation of the proposed approaches is somewhat limited, lacks proper comparisons with existing methods, which may raise doubt about the reliability and effectiveness of the proposed solution. A strong experimental validation is crucial to establish the viability of the proposed techniques.

**Reproducibility:**

4: Could mostly reproduce the results, but there may be some variation because of sample variance or minor variations in their interpretation of the protocol or method.

**Reviewer Confidence:**

3: Pretty sure, but there's a chance I missed something. Although I have a good feel for this area in general, I did not carefully check the paper's details, e.g., the math, experimental design, or novelty.

---

> ### Author Rebuttal · Authors · 2023-08-29
>
> Thank you for your thoughtful review comments. In the following response, we have organized our answers to address each point raised in your comments. We are committed to incorporating additional results in the future revision of the paper based on your feedback.
>
> **Limited choices of baseline methods**
>
> We acknowledge the reviewer's concern regarding the limited experimental evaluation and comparisons with existing methods. However, it's important to note that the scarcity of directly comparable baseline methods is precisely why our work focuses on this new direction. Previous works have either concentrated on benchmarking task performers (Figure 2 i) or on benchmarking helper-performer agent pairs (Figure 2 ii). Our work proposes a new benchmark specifically for helper agents (Figure 2 iii), filling a gap in the existing literature.
>
>
>
>
> **Additional baselines**
>
> We agree that comprehensive baselines are essential for validating our approach, particularly in light of the growing focus on Large Language Models (LLMs). Given the complexity of our tasks, the multimodal LLM (mPLUG-Owl) serves as an adequate LLM baseline as it inherently supports our task requirements by natively accommodating both image sequences and text for language generation. Additionally, we have also explored two additional baselines that leverage the power of LLMs:
>
> 1. Multimodal-LLM + ChatGPT: We also experimented with prompting the Multimodal-LLM (mPLUG-Owl) to generate only captions for the image sequences. Subsequently, we used ChatGPT to serve as the helper agent, generating responses based on the captions and queries from the task performer.
> 2. BLIP2 Model with stacked images input: We employed the BLIP2 model [1] in a zero-shot manner. BLIP2 model benefits from a generic and efficient pre-training strategy that combines pretrained vision models with LLMs for vision-language pretraining. Due to the model's limitation of accepting only a single image input, we stacked the input image sequences into a single image arranged in four columns. The text prompt used is the same as the one we adopted for the Multimodal-LLM (mPLUG-Owl) in Table 4 of our paper.
>
> The result of additional baselines is shown in the following table with the same format as in Table 1 from the paper.
>
> |                                         | CVDN validation seen GP | CVDN validation unseen GP | DialFRED validation seen SR | DialFRED validation unseen SR | AVDN validation seen SPL | AVDN validation unseen SPL |
> |-----------------------------------------|-------------------------|---------------------------|-----------------------------|-------------------------------|--------------------------|----------------------------|
> | Multimodal LLM baseline reported in the paper (mPLUG-Owl)     | 5.3                     | 3.6                       | 47.0                        | 33.8                          | 0.7                      | 1.5                        |
> | Additional baseline mPLUG-Owl + ChatGPT | 5.3                     | 3.5                       | 43.5                        | 31.8                          | 1.2                      | 1.6                        |
> | Additional baseline BLIP2               | 5.6                     | 3.4                       | 38.6                        | 24.7                          | 3.8                      | 5.0                        |
>
>
> Overall, our additional baseline mPLUG-Owl + ChatGPT shows performance comparable to the multimodal LLM baseline mPLUG-Owl reported in the paper. The BLIP2 baseline is weak on the DialFRED dataset but performs particularly well on the AVDN dataset.
>
>
> **Reference:**
> [1] Li, Junnan, et al. "Blip-2: Bootstrapping language-image pre-training with frozen image encoders and large language models." arXiv preprint arXiv:2301.12597 (2023).

---

### Official Review · Reviewer_yJ8D · 2023-08-06

**Soundness:** 3

**Excitement:**

4: Strong: This paper deepens the understanding of some phenomenon or lowers the barriers to an existing research direction.

**Paper Topic And Main Contributions:**

The paper introduces the Respond to Help (R2H) benchmark for developing intelligent navigation helper agents. It includes two tasks: Respond to Dialog History (RDH) and Respond during Interaction (RdI). The authors present two approaches, including a novel model named SeeRee, and a multi-modal large language model. Meanwhile the authors conducted experiments on existing vision-and-dialog navigation datasets to demonstrate the effects.

**Reasons To Accept:**

1. The R2H benchmark could become a standard for evaluating and comparing models in the domain of multi-modal navigation, fostering further research and innovation.
2. The paper explores two approaches to construct navigation helper agents: fine-tuning a task-oriented multi-modal response generation model named SeeRee, and employing a multi-modal large language model in a zero-shot manner.


**Reasons To Reject:**

1. The challenges related to real-world complexity and dataset scarcity may limit the broader applicability of the proposed model.
2. Authors may need to use other more LLMs for experimental comparisons.

**Reproducibility:**

3: Could reproduce the results with some difficulty. The settings of parameters are underspecified or subjectively determined; the training/evaluation data are not widely available.

**Reviewer Confidence:**

3: Pretty sure, but there's a chance I missed something. Although I have a good feel for this area in general, I did not carefully check the paper's details, e.g., the math, experimental design, or novelty.

---

> ### Author Rebuttal · Authors · 2023-08-29
>
> Thank you for your thoughtful review comments. In the following response, we have organized our answers to address each point raised in your comments. We are committed to incorporating additional results in the future revision of the paper based on your feedback.
>
> **Challenges in building helper agents**
>
> Due to the challenges posed by real-world complexity and dataset scarcity, the task of building helper agents becomes not just difficult but also critically important. It is precisely these challenges that underscore the urgent need for focused research in this area. Our work aims to advance this important direction by developing helper agents that can be beneficial in real-life applications.
>
>
> To mitigate the issue of data scarcity, we have leveraged three existing datasets that were originally developed for benchmarking task performers. By doing so, we not only make efficient use of available resources but also demonstrate the broader applicability of our model.
>
>
>
> Our benchmark is the first to focus specifically on the development of multimodal navigational helper agents. We believe that this will help to foster more future works in this direction, thereby gradually addressing the challenges related to real-world complexity and dataset scarcity.
>
>
>
> **Additional baselines**
>
> We agree that comprehensive baselines are essential for validating our approach, particularly in light of the growing focus on Large Language Models (LLMs). Given the complexity of our tasks, the multimodal LLM (mPLUG-Owl) serves as an adequate LLM baseline as it inherently supports our task requirements by natively accommodating both image sequences and text for language generation. Additionally, we have also explored two additional baselines that leverage the power of LLMs:
>
> 1. Multimodal-LLM + ChatGPT: We also experimented with prompting the Multimodal-LLM (mPLUG-Owl) to generate only captions for the image sequences. Subsequently, we used ChatGPT to serve as the helper agent, generating responses based on the captions and queries from the task performer.
> 2. BLIP2 Model with stacked images input: We employed the BLIP2 model [1] in a zero-shot manner. BLIP2 model benefits from a generic and efficient pre-training strategy that combines pretrained vision models with LLMs for vision-language pretraining. Due to the model's limitation of accepting only a single image input, we stacked the input image sequences into a single image arranged in four columns. The text prompt used is the same as the one we adopted for the Multimodal-LLM (mPLUG-Owl) in Table 4 of our paper.
>
> The result of additional baselines is shown in the following table with the same format as in Table 1 from the paper.
>
> |                                         | CVDN validation seen GP | CVDN validation unseen GP | DialFRED validation seen SR | DialFRED validation unseen SR | AVDN validation seen SPL | AVDN validation unseen SPL |
> |-----------------------------------------|-------------------------|---------------------------|-----------------------------|-------------------------------|--------------------------|----------------------------|
> | Multimodal LLM baseline reported in the paper (mPLUG-Owl)     | 5.3                     | 3.6                       | 47.0                        | 33.8                          | 0.7                      | 1.5                        |
> | Additional baseline mPLUG-Owl + ChatGPT | 5.3                     | 3.5                       | 43.5                        | 31.8                          | 1.2                      | 1.6                        |
> | Additional baseline BLIP2               | 5.6                     | 3.4                       | 38.6                        | 24.7                          | 3.8                      | 5.0                        |
>
>
> Overall, our additional baseline mPLUG-Owl + ChatGPT shows performance comparable to the multimodal LLM baseline mPLUG-Owl reported in the paper. The BLIP2 baseline is weak on the DialFRED dataset but performs particularly well on the AVDN dataset.
>
>
> **Reference:**
> [1] Li, Junnan, et al. "Blip-2: Bootstrapping language-image pre-training with frozen image encoders and large language models." arXiv preprint arXiv:2301.12597 (2023).

---

### Meta-Review · Area_Chair_Wazq · 2023-09-18

**Recommendation:** 5

**Metareview:**

This paper introduces a benchmark that is a valuable contribution in that it could/should become a standard for evaluating navigation agents on the more specific characteristics introduced here (responding to help, keeping track of history, cooperating with the helper). The insights and novel attention-mask schemes are also good contributions to building better multi-modal navigation agents. The authors provided detailed answers to alleviate most reviewer concerns, added additional baselines to compare to, and added more analysis of the datasets used to address questions and concerns of the reviewers. This paper, and the evaluation benchmark that comes with it, is a good contribution to the language/RL/navigation field.

---

### Decision · Program_Chairs · 2023-10-07

**Decision:**

Accept-Main

**Comment:**

This paper introduces a benchmark that is a valuable contribution in that it could/should become a standard for evaluating navigation agents on the more specific characteristics introduced here (responding to help, keeping track of history, cooperating with the helper). The insights and novel attention-mask schemes are also good contributions to building better multi-modal navigation agents. The authors provided detailed answers to alleviate most reviewer concerns, added additional baselines to compare to, and added more analysis of the datasets used to address questions and concerns of the reviewers. This paper, and the evaluation benchmark that comes with it, is a good contribution to the language/RL/navigation field.